# Anxiety and Depressive Symptoms, Frailty and Quality of Life in Atrial Fibrillation

**DOI:** 10.3390/ijerph20021066

**Published:** 2023-01-06

**Authors:** Katarzyna Lomper, Catherine Ross, Izabella Uchmanowicz

**Affiliations:** 1Department of Clinical Nursing, Medical University, K. Bartla 5, 51-616 Wroclaw, Poland; 2The Centre for Cardiovascular Health, School of Health and Social Care, Edinburgh Napier University, Edinburgh EH11 4BN, UK

**Keywords:** atrial fibrillation, frailty syndrome, quality of life, anxiety symptoms, depressive symptoms

## Abstract

Introduction: Symptoms of atrial fibrillation (AF) can significantly affect functioning in daily life and reduce patients’ quality of life (QoL). The severity and type of AF symptoms affects not only patient’s QoL, but can be a cause of the development of emotional and psychological disorders. In addition, frailty syndrome (FS) plays important role from the point of view of developing disability and dependence on others, as well as reducing QoL. Aim: To assess the symptoms of anxiety and depression, to evaluate the co-occurrence of frailty syndrome and the impact of these factors on the quality of life of patients with AF. Methods: The study used a Polish adaptation of the Arrhythmia-Specific questionnaire in Tachycardia and Arrhythmia part III (ASTA part III), the Tilburg Frailty Indicator (TFI) and the Hospital Anxiety Depression Scale (HADS). Results: Analysis showed that anxiety symptoms and depressive symptoms correlate significantly (*p* < 0.05) and positively with the physical (r = 0.24; *p* < 0.001, r = 0.29, *p* = 0.002, respectively), psychological (r = 0.34, *p* < 0.001, r = 0.49 *p* < 0.001, respectively) and total quality of life (r = 0.31, *p* = 0.001, r = 0.414; *p* < 0.001, respectively) ASTA III domains. A significant (*p* < 0.05) positive correlation was observed between the TFI total score and the physical (r = 0.34, *p* < 0.001), psychological (r = 0.36, *p* < 0.001) and overall quality of life (r = 0.38, *p* < 0.001) in ASTA III domains. Conclusions: Both FS and depressive and anxiety symptoms significantly affect QoL. Understanding the relationship between anxiety and depressive symptoms, FS and QoL may allow for a more targeted approach to the treatment and care of patients with AF.

## 1. Introduction

The prevalence of atrial fibrillation (AF) varies by geographic region and ethnicity, although old age is one of the main factors in the development of the arrhythmia [1]. Old age is also a risk factor for the development of frailty syndrome (FS) [2]. The association of AF with frailty has not been thoroughly documented; however, it has been shown that patients with AF are more likely to be diagnosed with FS [3]. The incidence of FS in the AF population varies strongly, ranging from 5.9% to 85.5%. Additionally, FS independently predicts all-cause mortality and major bleeding in patients with AF, which can further affect the functioning of patients [4].

Symptoms of atrial fibrillation can significantly affect functioning in daily life and reduce patients’ quality of life (HRQoL—Health Related Quality of Life) [5]. The impact of AF on a patient’s well-being depends mainly on the frequency of attacks, the duration of the disease and the severity of symptoms. Recurrent episodes of arrhythmia result in a loss of quality of life [6]. In addition, patients with AF have a worse QoL compared to healthy individuals [7], which is comparable to the quality of life of patients with severe cardiovascular disease and other arrhythmias [8]. The severity and type of AF symptoms affects not only patients’ QoL, but can also cause the development of emotional and psychological disorders. Anxiety and depressive symptoms have been shown to increase with the recurrence of arrhythmia episodes and are related to the severity of AF symptoms [9,10]. European Society of Cardiology (ESC) guidelines for the management of atrial fibrillation point to the possibility of mood disorders in this population [7]. Moreover, the presence of depressive and anxiety symptoms can affect the results of the therapeutic process in AF [11].

Early recognition of FS is important and can support the understanding of the development of disability and dependence on others, as well as reduction in quality of life [12,13]. The use of HRQoL assessments indicates the disease and treatment related limitations that patients experience in their daily lives [14], and therefore, may be helpful in choosing an appropriate treatment strategy and advising on lifestyle modifications. Few studies have evaluated or sought to explain the relationship between FS and QoL among patients with AF. 

Therefore, the aim of the study was to assess symptoms of anxiety and depression and to evaluate the co-occurrence of frailty syndrome and the impact of these factors on the quality of life for in patients with AF. 

## 2. Materials and Methods

### 2.1. Study Settings and Participants

The study was conducted in a hospital setting among 116 consecutively selected patients hospitalized for atrial fibrillation in a cardiology department in Wroclaw, Poland. Average age of the studied group was 75.2 years (SD = 8.2) The criteria for inclusion were: non-valvular atrial fibrillation, taking oral anticoagulation, no cognitive impairment preventing unassisted completion of the questionnaires and providing written and informed consent to participate in the study. The exclusion criteria for the study were any condition where the patient required intensive cardiac care and a history of stroke within the last 6 months.

Patients were asked to complete survey questionnaires accompanied by a nurse.

### 2.2. Ethical Considerations

The study was approved by the independent Bioethics Committee of the Wroclaw Medical University, Poland. All participants were informed about the purpose of the study, its conduct, and the possibility of withdrawal at any stage. The study was conducted in accordance with the requirements of the Declaration of Helsinki.

### 2.3. Research Tools

The study used a Polish adaptation of the Arrhythmia-Specific questionnaire in Tachycardia and Arrhythmia part III (ASTA part III) [15], the Tilburg Frailty Indicator (TFI) [16] and the Hospital Anxiety Depression Scale (HADS) [17].

The ASTA part III evaluates the influence of arrhythmia on the patient’s daily life, HRQoL, comprising 13 questions related to the impact of arrhythmia on daily physical (7 items) and mental (6 items) function. The ASTA HRQoL total scale score ranges from 0 (best possible HRQoL) to 39 (worst possible HRQoL). Higher scores reflect a more negative impact of arrhythmia on the HRQoL. The study used a Polish version adapted by Lomper et al. [15,18].

The TFI allows for a comprehensive and reliable evaluation of FS, including its physical, psychological and social components. The physical subscale (0–8 points) measures physical health, unintentional weight loss, difficulty in walking, balance, hearing and vision problems, grip strength, and physical fatigue. The psychological subscale includes such factors as memory problems, feeling down, feeling nervous or anxious, and inability to cope with problems. The social subscale includes three factors: living alone, lack of social relations, and lack of social support. Eleven items have two response categories, “yes” and “no”, and four items also have the response category “sometimes”. After recoding, score ranges are as follows: 0–15 (overall frailty), 0–8 (physical frailty), 0–4 (psychological frailty), and 0–3 (social frailty). The total TFI score ranges between 0 and 15 points, and scores over 5 points are considered diagnostic for FS [16]. The study used a Polish version adapted by Uchmanowicz et al. [19].

The HADS is a self-reported screening scale which measures anxiety as a state and not a trait [17]. It comprises two separate subscales, assessing anxiety and depression. Answers are given using a 4-item Likert scale, where the minimum score is 0, and the maximum is 21. Scores of 0–7 points indicate no anxiety or depression symptoms; 8–10 points are considered a borderline result, while 11–21 points indicate a marked disorder [20]. The study used a Polish version adapted by Majkowicz et al. [21]. 

### 2.4. Statistical Analysis

Analysis of quantitative variables (i.e., expressed by number) was carried out by calculating the mean, standard deviation, median and quartiles. Analysis of qualitative variables (i.e., not expressed by number) was carried out by calculating the number and percentage of occurrences of each value. Comparisons between quantitative variables in the two groups were made using the Mann–Whitney test. Correlations between quantitative variables were analyzed using Spearman’s correlation coefficient.

The analysis assumed a significance level of 0.05. All *p*-values below 0.05 were interpreted as indicating significant correlations. The analysis was performed in R software, version 4.1.3 [22].

## 3. Results

### 3.1. Sociodemographic and Clinical Data

An analysis of the socjodemographic and clinical data showed that the majority of the study group were men (55.17%) and urban residents (83.62%). Analysis of clinical data showed that 56.03% had permanent atrial fibrillation. Most of the respondents had the disease for up to 5 years (59.48%). The data is presented in Table 1.

### 3.2. Quality of Life (ASTA III Questionnaire)

An analysis of the ASTA III questionnaire showed that quality of life in the physical domain averaged 25.37 points (Median 23.81, Q1 9.52; Q3 38.1) and in the psychological domain averaged 27.92 points (Median 27.92, Q1 16.67; Q3 34.72). Total quality of life averaged 26.65 points (SD = 16.29). The data is presented in Table 2.

### 3.3. Analysis of Anxiety and Depression (HADS-M Questionnaire)

In the study group, on the anxiety scale, 77 out of 116 survey participants (66.38%) had no symptoms, 23 respondents (19.83%) had pronounced symptoms and 16 respondents (13.79%) had a borderline condition. On the depression scale; 93 of 116 survey participants (80.17%) had no symptoms, 13 respondents (11.21%) had a borderline condition, and 10 respondents (8.62%) had pronounced symptoms. The data is presented in Table 3.

### 3.4. Analysis of the Occurrence of Frailty Syndrome (TFI Questionnaire)

Analysis of the TFI questionnaire showed the presence of FS in 78 patients (67.24%). The data is presented in Table 4.

### 3.5. Correlation between Quality of Life (ASTA III) and Anxiety and Depression Disorders (HADS-M)

Analysis demonstrated that anxiety symptoms correlate significantly (*p* ˂ 0.05) and positively (r ˃ 0) with the physical (*p* = 0.009), psychological (*p* < 0.001) and total (*p* = 0.001) quality of life ASTA III domains. Therefore, the greater the severity of anxiety symptoms the greater the limitations in these areas, although the correlation can be considered weak. The data is shown in Table 5. 

It was also noted that depressive disorders correlate significantly (*p* ˂ 0.05) and positively with the physical (*p* = 0.002), psychological (*p* < 0.001) and total quality of life (*p* < 0.001) ASTA III domains, so the greater the severity of depression symptoms, the greater the limitations in these areas. The data is shown in Table 5.

### 3.6. Correlation between Quality of Life (ASTA) and Frailty Syndrome (TFI)

A significant (*p* ˂ 0.05), positive correlation was observed between the TFI total score and the physical (*p* < 0.001), psychological (*p* < 0.001) and total quality of life (*p* < 0.001) in ASTA III domains; therefore, the greater the severity of frailty syndrome, the greater the limitations in these areas. The data is shown in Table 6.

## 4. Discussion

Symptoms and severity of atrial fibrillation can affect the onset of disability and determine the patient’s quality of life [6]. It has also been shown that AF patients have worse QoL regardless of disease symptoms compared to patients with other cardiovascular diseases [8]. The factors influencing HRQoL in AF are not fully understood. Studies demonstrated different results in the influence of socio-demographic factors, such as age or gender, and also of clinical factors including comorbidities, treatment or the presence of depression and anxiety symptoms [23].

Recommendations for AF therapy articulate the need to reduce the severity of symptoms and prevent complications from the arrhythmia. The use of an appropriate AF treatment strategy is essential to improve QoL [6]. Clinical indicators are important in assessing the effectiveness of treatment, but are currently not sufficient, and it is therefore recommended to assess HRQoL to take into account patient self-assessment [14].

Clinical trials on intervention treatment strategies (rate or rhythm control) demonstrated lower HRQoL values for patients with AF compared to the general population or control groups [24,25]. The effect of the use of ablation [26] and cardioversion [27] on the sense of quality of life in AF has also been documented. 

However, there is limited data that focuses on assessing the impact of arrhythmias on daily functioning in the general population of patients with AF and the available studies mostly use generic tools to assess QoL. The ASTA HRQoL questionnaire used in our study was designed to assess both the burden of arrhythmia-specific symptoms and their impact on HRQoL and daily functioning [15,18]. In our own study, the highest values, and therefore the biggest impact of AF symptoms on subjective quality of life were observed in the psychological domain, total quality of life and the physical domain. In comparison, Charitakis et al. in a study examining factors influencing arrhythmia-related symptoms and HRQoL in patients with AF reported higher quality of life scores on the ASTA questionnaire; for the overall scale (36 points), for the physical scale (38 points), and for the mental scale (28 points). This indicates an even higher burden of disease symptoms, and thus a lower rated quality of life [28]. We can therefore conclude that there is a poor quality of life in this group of patients. 

A number of factors, including symptoms of the disease, may contribute to lower quality of life of patients with AF. The most commonly reported symptoms of AF are palpitations, shortness of breath during activity, fatigue and anxiety [29]. 

Such symptoms often cause AF patients to experience psychological stress, which may manifest as anxiety or depressive symptoms, leading to increased mortality and increased hospitalizations [30]. Moreover, anxiety and depressive symptoms may worsen with each new episode of arrhythmia [9] and are associated with increased AF symptoms and thus lower quality of life [10]. 

A cross-sectional studies of 116 patients with AF indicates a significant relationship between disease symptom burden and psychological distress vs. sense of HRQoL [31]. We obtained similar results in the present study. We showed a significant relationship between anxiety and depressive symptoms and HRQoL. Higher levels of anxiety symptoms resulted in the deterioration of quality of life in both physical and psychological domains, as well as overall quality of life. We noted similar results in the context of depressive symptoms. Higher severity of depressive symptoms negatively influenced the quality of life in patients with AF.

Patients with AF are more likely to be diagnosed with frailty syndrome. As previously mentioned, the prevalence of FS in the AF population varies widely [4]. In this study, based on the TFI questionnaire, frailty syndrome was diagnosed in 67.24% of patients. Importantly, the co-occurrence of FS among patients with AF may be responsible for a number of adverse sequelae, including lack of or inadequate treatment with oral anticoagulants (OACs), increased risk of stroke, and increased mortality rates [32]. Published studies found lower patient satisfaction during oral anticoagulant treatment in a population with AF with comorbid frailty syndrome, an association between the presence of anxiety and depression symptoms and the occurrence of FS [33] and an association between the presence of anxiety and depressive symptoms and the presence of FS [34]. 

There is little known about the effect of FS on HRQoL in AF [35]. Slawuta et al. conducted an analysis of the effect of FS on HRQoL on a group of 158 patients with AF using the Edmonton Frail Scale (EFS) and ASTA HRQoL questionnaires. In the regression analysis frailty syndrome was a significant and independent factor in arrhythmia symptom severity and worsened quality of life [35]. Likewise, in this study we observed a negative impact of FS on quality of life. The co-occurrence of FS was associated with greater limitations in overall quality of life (r = 0.38), in the psychological (r = 0.36) and the physical (r = 0.34) domains. Both the prevalence of FS and AF are significantly associated with old age. Overlapping intractable disease symptoms and FS-related changes may be responsible for reduced quality of life. In addition, FS has similar predictive power for unplanned hospitalizations, hemorrhagic complications and mortality compared to the CHA2DS2-VASc score used to assess embolic risk in AF [36]. Understanding the relationship between frailty and quality of life may allow for a more targeted approach to the treatment and care of patients with AF and ultimately lead to better treatment outcomes in this group of patients. It is claimed that frailty can be managed in the hospital with interventions such as physiotherapy [37], nutritional therapy [38] and comprehensive geriatric care (CGA) [39].

A low sense of quality of life can affect a patient’s daily functioning, but it can also be a cause of non-adherence to treatment recommendations, and therefore increased hospitalizations and deaths, with negative consequences for the health care system. 

## 5. Conclusions

FS and depressive and anxiety symptoms significantly affect QoL. Understanding the relationship between anxiety and depression symptoms, FS and QoL may allow for a more targeted approach to the treatment and care of patients with AF. Due to the risk of consequences which may in part be irreversible, screening for FS and anxiety and depression symptoms in AF is recommended. In patients diagnosed with FS, it is worthwhile to involve the family members or caregivers in the therapeutic process. 

There are studies that indicates that depressive symptoms are closely related to the occurrence and development of AF, which may also increase the complexity of management in AF patients [40].

## 6. Implication for Practice

Both atrial fibrillation and frailty syndrome may affect patients’ sense of quality of life. Moreover, FS can have a significant impact on clinical decision-making in patients with AF. The available literature increasingly identifies frailty as a factor, prioritized over chronological age as more useful for therapeutic decision-making in AF patients. Assessment of anxiety and depressive disorders in AF patients should be the foundation for developing and implementing appropriate therapeutic interventions.

The co-occurrence of intractable atrial fibrillation symptoms with frailty syndrome and symptoms of anxiety and depression can significantly worsen the HRQoL. Therefore, it is advisable to routinely assess the quality of life itself, but also to analyze the factors that have a negative impact on the HRQoL. Therapeutic teams, which should include doctors, nurses and psychologists, should take measures aimed at improving the QoL of AF patients. 

## 7. Limitations

A limitation of the study is the relatively small number of patients subjected to the questionnaire survey. Another limitation is that this was a single-center study. The study did not evaluate the impact of other sociodemographic and clinical factors that may affect the sense of quality of life and the occurrence of anxiety and depression symptoms in the AF patients.

## Figures and Tables

**Table 1 ijerph-20-01066-t001:** Sociodemographic and clinical data.

Parameter		Total (%)	Total (Number)
Sex	Female	55.7%	64
	Male	44.83%	52
Place of residence	Village	16.38%	19
	City	83.62%	97
Education	Primary	25.00%	29
	Professional training	22.41%	26
	Secondary	37.93%	44
	Higher	14.66%	17
Marital status	In relationship	52.59%	61
	Single	47.41%	55
Type of AF	Paroxysmal/Persistent	43.97%	51
	Permanent	56.03%	65
Duration of illness	Up to 5 years	59.48%	69
	6–10 years	18.97%	22
	Over 10 years	21.55%	25
EHRA Classification	EHRA 1	12.93%	15
	EHRA 2	36.21%	42
	EHRA 3	44.83%	52
	EHRA 4	6.03%	7

AF—atrial fibrillation; EHRA—European Heart Rhythm Association Classification.

**Table 2 ijerph-20-01066-t002:** Quality of life (ASTA III questionnaire).

ASTA III—Quality of Life [Points]	Mean	SD	Median	Min	Max	Q1	Q3
Physical domain	25.37	19.31	23.81	0	85.71	9.52	38.1
Psychological domain	27.92	16.79	27.78	0	83.33	16.67	34.72
Overal quality of life	26.65	16.29	25.99	0	82.14	13.69	37.8

ASTA—the Arrhythmia-Specific questionnaire in Tachycardia and Arrhythmia; SD—standard deviation; Min—minimum; Max—maximum; Q1—first quartile; Q3—third quartile.

**Table 3 ijerph-20-01066-t003:** Analysis of the prevalence of anxiety and depressive symptoms (HADS-M questionnaire).

Intensity	HADS	
	Anxiety Symptoms (Total%)	Depression Symptoms (Total%)
No symptoms	77 (66.38%)	93 (80.17%)
Borderline condition	16 (13.79%)	13 (11.21%)
Pronounced symptoms	23 (19.83%)	10 (8.62%)

HADS—Hospital Anxiety and Depression Scale.

**Table 4 ijerph-20-01066-t004:** Analysis of the occurrence of frailty syndrome (TFI questionnaire).

TFI—Points	Interpretation	Total (%)	Total (Number)
0–4	Without frailty syndrome	32.76%	33
5 and more	Frailty syndrome	67.24%	78

TFI—Tilburg Frailty Indicator.

**Table 5 ijerph-20-01066-t005:** Correlation between quality of life (ASTA III) and anxiety and depression symptoms (HADS-M).

ASTA III—Quality of Life [Points]	Anxiety Symptoms
	Spearman Rank Correlation
Physical domain	r = 0.241, *p* = 0.009 *
Psychological domain	r = 0.344, *p* < 0.001 *
Overall quality of life	r = 0.31, *p* = 0.001 *
**ASTA III—Quality of Life [Points]**	**Depression Symptomos**
	**Spearman Rank Correlation**
Physical domain	r = 0.289, *p* = 0.002 *
Psychological domain	r = 0.486, *p* < 0.001 *
Overall quality of life	r = 0.414, *p* < 0.001 *

* statistically significant relationship (*p* < 0,05); r—Spearman’s correlation coefficient; ASTA—the Arrhythmia-Specific questionnaire in Tachycardia and Arrhythmia.

**Table 6 ijerph-20-01066-t006:** Correlation between quality of life (ASTA III) and frailty syndrome (TFI).

ASTA III—Quality of Life [Points]	TFI Total Score
	Spearman Rank Correlation
Physical domain	r = 0.336, *p* < 0.001 *
Psychological domain	r = 0.355, *p* < 0.001 *
Overall quality of life	r = 0.383, *p* < 0.001 *

* statistically significant relationship (*p* < 0,05); r—Spearman’s correlation coefficient; ASTA—the Arrhythmia-Specific questionnaire in Tachycardia and Arrhythmia; TFI—Tilburg Frailty Indicator.

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
