# Peer review of "Anxiety and Depressive Symptoms, Frailty and Quality of Life in Atrial Fibrillation"

_ijerph, 2023, doi:10.3390/ijerph20021066_

Round 1

Reviewer 1 Report

The study aimed to assess symptoms of anxiety and depression with co-occurrence of frailty syndrome and the impact of those factors on the quality of life in patients with atrial fibrillation.

In general, this is a very well-written manuscript, with every aspect of the study design adequately discussed. I would only suggest minor adjustments like:

- adding Ethics Committee approve number

- line 106 and 257 – missing text

Moreover, I would suggest adding some visualisation of how examined parameters (ASTA and HRQOL, HADS, TFI) interact with each other. In other words, how those results are connected and which one is connected to which.

Author Response

Author's Reply to the Review Report (Reviewer 1)

The study aimed to assess symptoms of anxiety and depression with co-occurrence of frailty syndrome and the impact of those factors on the quality of life in patients with atrial fibrillation.

In general, this is a very well-written manuscript, with every aspect of the study design adequately discussed. I would only suggest minor adjustments like:

- adding Ethics Committee approve number

In accordance with the reviewer's comment, we have added the bioethics committee number

- line 106 and 257 – missing text

The missing text is perhaps a technical error. We checked the manuscript and line 105-106 is the following sentence: The HADS is a self-reported screening scale which measures anxiety as a state and not a trait. [17] It comprises two separate subscales, assessing anxiety and depression.

We checked the manuscript and line 256-257 is the following sentence: Therepautic teams, which should include doctors, nurses and psychologists, should take measures aimed at improving the QoL of AF.

We hope that now the excerpt is visible

Moreover, I would suggest adding some visualisation of how examined parameters (ASTA and HRQOL, HADS, TFI) interact with each other. In other words, how those results are connected and which one is connected to which.

In accordance with the reviewer's comment, we performed additional analysis, the results of which are attached below. All variables included in the analysis correlated with each other statistically significantly, except for HADS anxiety with TFI. This was not an area of our research, although if the Reviewer deems it advisable we can include the results presented in the paper. Thank you.

HADS: Anxiety

HADS: Depression

ASTA III: Physical domain

ASTA III: Psychological domain

ASTA III: Overall quality of life

TFI: Total

HADS: Anxiety

---

r=0,752, p<0,001 *

r=0,241, p=0,009 *

r=0,344, p<0,001 *

r=0,31, p=0,001 *

r=0,154, p=0,099

HADS: Depression

r=0,752, p<0,001 *

---

r=0,289, p=0,002 *

r=0,486, p<0,001 *

r=0,414, p<0,001 *

r=0,336, p<0,001 *

ASTA III: Physical domain

r=0,241, p=0,009 *

r=0,289, p=0,002 *

---

r=0,636, p<0,001 *

r=0,908, p<0,001 *

r=0,336, p<0,001 *

ASTA III: Psychological domain

r=0,344, p<0,001 *

r=0,486, p<0,001 *

r=0,636, p<0,001 *

---

r=0,89, p<0,001 *

r=0,355, p<0,001 *

ASTA III: Overall quality of life

r=0,31, p=0,001 *

r=0,414, p<0,001 *

r=0,908, p<0,001 *

r=0,89, p<0,001 *

---

r=0,383, p<0,001 *

TFI: Total

r=0,154, p=0,099

r=0,336, p<0,001 *

r=0,336, p<0,001 *

r=0,355, p<0,001 *

r=0,383, p<0,001 *

---

r – Spearman rank correlation

* statistically significant relationship (p<0,05)

The correlation coefficients between the analyzed variables are shown in the chart below (the so-called heat map). Blue areas indicate positive correlations (correlation coefficient (r) greater than 0), and red areas indicate negative correlations (r < 0). White areas represent no correlation.

We want to thank again very much for Reviewers time and valuable suggestions for our paper. 

Reviewer 2 Report

Dear Author/s,

I appreciate the enormous amount of work you have contributed to the submitted article.

The research makes an contribution to the literature. After taking into account some significant fixes.

1.      I suggest that the title needs improvement, unjustified - questionnaire-based study.

2.      Introduction -  the recommended additions / extensions to the subject to a medium degree.

3.      Material and methods. 

It must be supplemented. Study settings and Participants / General Characteristics

Were the nurses members of the research team in the project?

What tool was used to assess cognitive function? Please introduce.

Did the patients fill in the questions themselves, or did the nurse read the questions?

4.      Results

Demographic and social characteristics of the study group general information - none (e.g. gender).

Is it reasonable to enter N in Table 1? Below the table there is no legend of abbreviations appearing in the table.

Table 2. Necessary correction of the presented results N (%).

5.      The discussion is well written. I suggest that the authors separate the limitations from the strengths of the study.

Author Response

Author's Reply to the Review Report (Reviewer 2)

Dear Author/s,

I appreciate the enormous amount of work you have contributed to the submitted article.

The research makes an contribution to the literature. After taking into account some significant fixes.

  1. I suggest that the title needs improvement, unjustified - questionnaire-based study.

In accordance with the reviewer's comment, the title of the work was corrected

  1. Introduction -  the recommended additions / extensions to the subject to a medium degree.
  2. Material and methods. 

It must be supplemented. Study settings and Participants / General Characteristics - in accordance with the reviewer's comment, we have supplemented the mauscript with the general characteristics of the patients (table number 1)

Were the nurses members of the research team in the project? Patients completed the research questionnaires in the presence of a nurse who was a member of the team and  the author of the manuscript.

What tool was used to assess cognitive function? Please introduce. - To assess cognitive function, we used the MMSE screening tool, which is a commonly used screening tool.

Did the patients fill in the questions themselves, or did the nurse read the questions? - Patients who were qualified for the study completed the survey questionnaires independently, in the presence of the research nurse

  1. Results

Demographic and social characteristics of the study group general information - none (e.g. gender). in accordance with the reviewer's comment, we have supplemented the mauscript with the general characteristics of the patients (table number 1)

Is it reasonable to enter N in Table 1? Below the table there is no legend of abbreviations appearing in the table.

Table 2. Necessary correction of the presented results N (%).

In accordance with the reviewer's remark, the tables have been reworded. We removed the numerical values of "N". The legend under the tables was completed.

  1. The discussion is well written. I suggest that the authors separate the limitations from the strengths of the study.

In accordance with the reviewer's comment, the limitation section was written separately. We want to thank for insightful analysis of our paper and valuable guidance.

We want to thank again very much for Reviewers time and valuable suggestions for our paper. 

Author's Reply to the Review Report (Reviewer 3)

  1. in the method section it was stated that the study was conductedin a hospital setting among 116 patients hospitalized for atrial fibrillation. It should be added during which period patients were recruited and in which hospital or department. How these 116 patients were chosen for the study? Were they consecutive patients who were admitted in the hospital during observed period?

We want to thank The Reviewer for insightful analysis of our paper. We included consecutively hospitalized patients within the cardiology department who met the inclusion criteria for the study. The methodology section is corrected.

  1. How diagnosis of atrial fibrillation was defined?

Patients recruited for the study had a previous diagnosis of atrial fibrillation; classification was obtained from analysis of medical records.

  1. Taking into account that patients were asked to complete several questionnaires it would be useful to know did any of the patients refused to participate in the study?

We want to thank The Reviewer  for the relevant question, however none of the eligible patients refused to participate in the study.

  1. Also there is no data about study subjects. We do not know age and gender of these patients? It should be present in the paper these main characteristic of the patients? Especially because quality of life in these patients depends on age and gender. It also should be discussed.

We want to thank The Reviewer  for the valuable tip. We have supplemented the paper with sociodemographic and clinical data, which we have included in table number 1

  1. It was mentioned in the results section: “Analysis demonstrated that anxiety symptoms correlate significantly (pË‚0.05) and positively (r˃0) with the physical (p=0,009), psychological (p<0,001) and total (p=0,001) quality of life ASTA III domains, therefore the greater the severity of anxiety symptoms the greater the limitations in these areas“.However, also it should be mentioned are these correlations strong or weak for all correlations in the study.

We have made corrections in accordance with the Reviewer's comments

  1. The quality of life, and also levels of depression and anxiety depend not only on presence of atiral fibrillation. It is connected with many other factors. It should be explained in the study limitations.

This is a very valuable comment, for which we want to thank. The limitation section of the paper has been revised as suggested by the Review

  1. Fix technical things in lines 105-112, then 176-177, in Table 2, actually whole paper must be checked for technical issues.

We want to thank for very insightful analysis of our paper. We hope that with this round of review there will be no technical issues

Reviewer 3 Report

1.  In the method section it was stated that the study was conducted in a hospital setting among 116 patients hospitalized for atrial fibrillation. It should be added during which period patients were recruited and in which hospital or department. How these 116 patients were chosen for the study? Were they consecutive patients who were admitted in the hospital during observed period?

2.      How diagnosis of atrial fibrillation was defined?

3.      Taking into account that patients were asked to complete several questionnaires it would be useful to know did any of the patients refused to participate in the study?

      How sample size was calculated?

4.      Also there is no data about study subjects. We do not know age and gender of these patients? It should be present in the paper these main characteristic of the patients? Especially because quality of life in these patients depends on age and gender. It also should be discussed.

5.      It was mentioned in the results section: “Analysis demonstrated that anxiety symptoms correlate significantly (pË‚0.05) and positively (r˃0) with the physical (p=0,009), psychological (p<0,001) and total (p=0,001) quality of life ASTA III domains, therefore the greater the severity of anxiety symptoms the greater the limitations in these areas“. However, also it should be mentioned are these correlations strong or weak for all correlations in the study.

6.      The quality of life, and also levels of depression and anxiety depend not only on presence of atiral fibrillation. It is connected with many other factors. It should be explained in the study limitations.

7.      Fix technical things in lines 105-112, then 176-177, in Table 2, actually whole paper must be checked for technical issues.

Author Response

Author's Reply to the Review Report (Reviewer 3)

  1. in the method section it was stated that the study was conductedin a hospital setting among 116 patients hospitalized for atrial fibrillation. It should be added during which period patients were recruited and in which hospital or department. How these 116 patients were chosen for the study? Were they consecutive patients who were admitted in the hospital during observed period?

We want to thank The Reviewer for insightful analysis of our paper. We included consecutively hospitalized patients within the cardiology department who met the inclusion criteria for the study. The methodology section is corrected.

  1. How diagnosis of atrial fibrillation was defined?

Patients recruited for the study had a previous diagnosis of atrial fibrillation; classification was obtained from analysis of medical records.

  1. Taking into account that patients were asked to complete several questionnaires it would be useful to know did any of the patients refused to participate in the study?

We want to thank The Reviewer  for the relevant question, however none of the eligible patients refused to participate in the study.

  1. Also there is no data about study subjects. We do not know age and gender of these patients? It should be present in the paper these main characteristic of the patients? Especially because quality of life in these patients depends on age and gender. It also should be discussed.

We want to thank The Reviewer  for the valuable tip. We have supplemented the paper with sociodemographic and clinical data, which we have included in table number 1

  1. It was mentioned in the results section: “Analysis demonstrated that anxiety symptoms correlate significantly (pË‚0.05) and positively (r˃0) with the physical (p=0,009), psychological (p<0,001) and total (p=0,001) quality of life ASTA III domains, therefore the greater the severity of anxiety symptoms the greater the limitations in these areas“.However, also it should be mentioned are these correlations strong or weak for all correlations in the study.

We have made corrections in accordance with the Reviewer's comments

  1. The quality of life, and also levels of depression and anxiety depend not only on presence of atiral fibrillation. It is connected with many other factors. It should be explained in the study limitations.

This is a very valuable comment, for which we want to thank. The limitation section of the paper has been revised as suggested by the Review

  1. Fix technical things in lines 105-112, then 176-177, in Table 2, actually whole paper must be checked for technical issues.

We want to thank for very insightful analysis of our paper. We hope that with this round of review there will be no technical issues.

Round 2

Reviewer 2 Report

1.      Material and methods. 

The region of Poland where the study was carried out is missing. There are no branches.

What tool was used to assess cognitive function? Please introduce.

2.      Results

Table 1, 3, 4  must be improved.

It must be N (i.e. the number of people tested / number) and %, e.g. 78 (36%).

Table 5, 6

What is "r"? There is no description or explanation in the "Statistical analysis" and under the tables.

The results presented in this version raise a lot of my doubts. Description not clear.

Author Response

We want to thank The Reviewer for insightful analysis of our paper.

  1. Material and methods. 

The region of Poland where the study was carried out is missing. There are no branches.

In accordance with the reviewer's comment, we have included information on where the research was conducted Wroclaw, Poland.

What tool was used to assess cognitive function? Please introduce.

To assess cognitive function, we used the MMSE screening tool, which is a commonly used screening tool.

  1. Results

Table 1, 3, 4  must be improved.

It must be N (i.e. the number of people tested / number) and %, e.g. 78 (36%).

Table 5, 6

What is "r"? There is no description or explanation in the "Statistical analysis" and under the tables.

The results presented in this version raise a lot of my doubts. Description not clear.

We have removed numerical values from the tables because this was suggested by one of the Reviewers. However, according to the comment, we have restored the tables to the original version, which we believe is also more readable. Thank you for this comment. Below the tables, we have included an explanation of the abbreviation "r", which was missing from the previous version of the paper.

Reviewer 3 Report

The manuscript is much improved.

Author Response

The manuscript is much improved.

We want to thank The Reviewer for insightful analysis of our paper.
